# Weakly Supervised Fine-grained Scene Graph Generation via Large Language Model

## Abstract

Weakly-Supervised Scene Graph Generation (WSSGG) research has recently emerged as an alternative to the fully-supervised approach that heavily relies on costly annotations. In this regard, studies on WSSGG have utilized image captions to obtain unlocalized triplets while primarily focusing on grounding the unlocalized triplets over image regions. However, they have overlooked the two issues involved in the triplet formation process from the captions: 1) *Semantic over-simplification* issue arises when extracting triplets from captions, where fine-grained predicates in captions are undesirably converted into coarse-grained predicates, resulting in a long-tailed predicate distribution, and 2) *Low-density scene graph* issue arises when aligning the triplets in the caption with entity/predicate classes of interest, where many triplets are discarded and not used in training, leading to insufficient supervision. To tackle the two issues, we propose a new approach, i.e., **L**arge **L**anguage **M**odel for weakly-supervised **SGG** (LLM4SGG), where we mitigate the two issues by leveraging the LLM's in-depth understanding of language and reasoning ability during the extraction of triplets from captions and alignment of entity/predicate classes with target data. To further engage the LLM in these processes, we adopt the idea of Chain-of-Thought and the in-context few-shot learning strategy. To validate the effectiveness of LLM4SGG, we conduct extensive experiments on Visual Genome and GQA datasets, showing significant improvements in both Recall@K and mean Recall@K compared to the state-of-the-art WSSGG methods. A further appeal is that LLM4SGG is data-efficient, enabling effective model training with a small amount of training images. Our code is available on https://anonymous.4open.science/r/LLM4SGG-83DD

## 1 Introduction

Scene Graph Generation (SGG) is a fundamental task in computer vision, aiming at extracting structured visual knowledge from images (Shi et al., 2019; Teney et al., 2017; Li et al., 2020; Wang et al., 2020; Yang et al., 2019; Zhong et al., 2020). Most existing SGG methods are fully-supervised, i.e., they heavily rely on the ground-truth annotations that involve the class information of entities and predicates as well as the bounding box of entities (Zellers et al., 2018; Yoon et al., 2023; Li et al., 2021b). However, since creating extensively annotated scene graph datasets is costly, the heavy reliance on these annotations imposes practical limitations on the model training (Zhang et al., 2023). To mitigate the high cost associated with manual annotations, weakly-supervised scene graph generation (WSSGG) approaches have recently emerged, aiming at training an SGG model without any annotated scene graph dataset. Specifically, the main idea of recent WSSGG methods is to leverage image captions along with associated images, as they can be easily collected from the Web (Li et al., 2022b; Zhang et al., 2023; Ye & Kovashka, 2021; Zhong et al., 2021).

The training process of WSSGG model using image captions requires four steps as illustrated in Figure 1(a). **Step 1**: *Preparing an image and its caption.* **Step 2**: *Parsing the image caption,* i.e., triplets formed as ⟨*subject, predicate, object*⟩ are extracted from the image caption through an off-the-shelf parser (Wu et al., 2019; Schuster et al., 2015). **Step 3**: *Aligning the triplets in the caption with entity/predicate classes of interest,* i.e., entity (subject, object) and predicate classes in the extracted triplets obtained in Step 2 are aligned with the entity and predicate classes in the target data[1], respectively. This alignment is based on their synonym/hypernym/hyponym contained in an external knowledge base (KB), e.g., WordNet (Miller, 1995). **Step 4**: *Grounding unlocalized*

---

[1] We use Visual Genome (Krishna et al., 2017) as the target data.

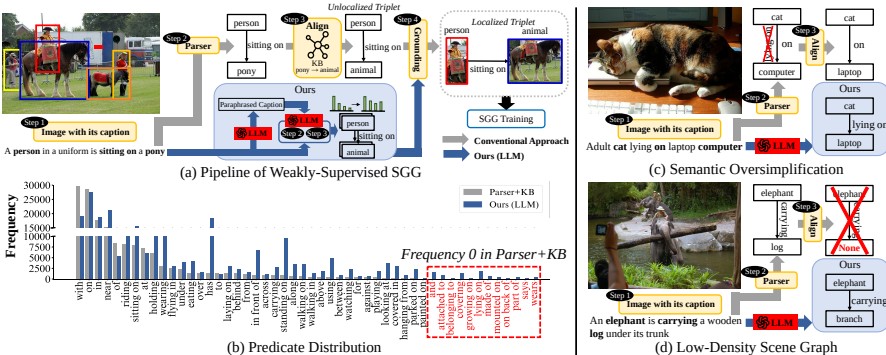

Figure 1: (a) The pipeline of weakly-supervised SGG. (b) The predicate distribution of unlocalized triplets (Parser+KB vs. Ours). In Parser+KB, the distribution becomes heavily long-tailed, and 12 out of 50 predicates are non-existent. (c) Semantic over-simplification caused by a rule-based parser in Step 2. (d) Low-density scene graph caused by the static structured of KB in Step 3.

*entities in the extracted triplets,* i.e., unlocalized entities (subjects and objects) are matched with relevant image regions generated by a pre-trained object detector, e.g., Faster R-CNN (Ren et al., 2015). The localized entities and predicates in the extracted triplets then serve as pseudo-labels for training an SGG model.

Existing WSSGG approaches mainly focus on Step 4 (Li et al., 2022b; Ye & Kovashka, 2021; Shi et al., 2021; Zhong et al., 2021). For example, in Figure 1(a), their efforts have been focused on grounding the entity person in an unlocalized triplet with an image region that captures the sitting behavior. More precisely, LSWS (Ye & Kovashka, 2021) exploits the contextual object information to accurately ground the unlocalized entities, leveraging the linguistic structure embedded within the triplets. Another line of research (Li et al., 2022b) employs a pre-trained vision-language model (Li et al., 2021a) to reflect the semantic interactions among entities within the image caption.

However, we argue that existing WSSGG approaches overlook the importance of the triplet formation process conducted in Step 2 and Step 3. We identify the two major issues described below, i.e., semantic over-simplication and low-density scene graph, which incur incomplete unlocalized triplets after Step 2 and 3. These incomplete triplets are mostly uninformative predicates with a limited number, and negatively impact the training of an SGG model even when entities are correctly grounded in Step 4. To demonstrate the impact of incomplete unlocalized triplets, we follow the conventional process to extract unlocalized triplets (i.e., Step 1-3), and conduct an examination of triplets obtained from COCO caption dataset, which are generated through Scene Parser (Wu et al., 2019) in Step 2 and WordNet (Miller, 1995) in Step 3. As a result, we identify the following two issues:

- **Semantic Over-simplification:** We find that the standard scene graph parser (Wu et al., 2019) operating on heuristic rule-based principles commonly used in Step 2 leads to a semantic over-simplification of predicates in extracted triplets. In other words, fine-grained predicates are undesirably converted into coarse-grained predicates, which we refer to as semantic over-simplification. For example, in Figure 1(c), an informative predicate lying on (i.e., fine-grained predicate) in the image caption is converted into a less informative predicate on (i.e., coarse-grained predicate), because the rule-based parser fails to capture the predicate lying on at once, and its heuristic rules fall short of accommodating the diverse range of caption's structure. As a result, the predicate distribution becomes heavily long-tailed, in which coarse-grained predicates (e.g., with, on, in) greatly outnumber fine-grained predicates (e.g., parked on, covered in) (Figure 1(b)). To make the matter worse, numerous fine-grained predicates eventually end up in a frequency of 0, even though they are originally present in the captions. Specifically, 12 out of 50 predicates are non-existent, which means that these 12 predicates can never be predicted, since the model is not trained on these predicates at all.

- **Low-Density Scene Graph:** We find that the KB-based triplet alignment in Step 3 leads to low-density scene graphs, i.e., the number of remaining triplets after Step 3 is small. We attribute the low-density scene graphs primarily to the utilization of KB in Step 3. Specifically, a triplet is discarded if any of the three components (i.e., subject, predicate, object) or their synonym/hypernym/hyponym within the triplet fail to align with the entity or predicate classes in the target data. For example, in Figure 1(d), the triplet ⟨elephant, carrying, log⟩ is discarded because

log does not exist in Visual Genome dataset nor its synonym/hypernym, even if elephant and carrying do exist. In Table 1, we report the number of triplets and images in Visual Genome dataset, which is a common benchmark dataset used in fully-supervised SGG approaches, and COCO caption dataset, which is a common benchmark dataset used in weakly-supervised SGG approaches. We observe that on average Visual Genome dataset contains 7.1 triplets (i.e., 405K/57K) per image (See Table 1(a)), while COCO dataset contains only 2.4 triplets (i.e., 154K/64K) per image (See Table 1(b)). This indicates that existing WSSGG approaches suffer from the lack of sufficient supervision per image, leading to poor generalization and performance degradation (Ye & Kovashka, 2021; Zareian et al., 2020). In summary, relying on the static structured knowledge of KB is insufficient to cover the semantic relationships among a wide a range of words, which incurs the low-density scene graph after Step 3.

To alleviate the semantic over-simplification and the low-density scene graph issues, we propose a new approach, namely **L**arge **L**anguage **M**odel for weakly-supervised **SGG** (LLM4SGG) that adopts a pre-trained Large Language Model (LLM), which has shown remarkable transferability to various downstream tasks in NLP such as symbolic reasoning, arithmetic, and common-sense reasoning (Touvron et al., 2023; Brown et al., 2020; Chowdhery et al.,

Table 1: Comparison of scene graph density.

| Dataset | How to annotate | # Triplet | # Image |
|---|---|---|---|
| **Fully-Supervised approach** | | | |
| (a) Visual Genome | Manual | 405K | 57K |
| **Weakly-Supervised approach** | | | |
| (b) COCO Caption | Parser+KB | 154K | 64K |
| (c) COCO Caption | LLM | 344K | 64K |

2022). Inspired by the idea of Chain-of-Thought[2] (CoT) (Wei et al., 2022b), which arrives at an answer in a stepwise manner, we separate the triplet formation process into two chains, each of which replaces the rule-based parser in Step 2 (i.e., Chain-1) and the KB in Step 3 (i.e., Chain-2). More precisely, we design a prompt for extracting triplets from a caption, and ask the LLM to extract triplets formed as <*subject, predicate, object*> (**Chain-1**). We expect that the predicates extracted based on a comprehensive understanding of the caption's context via LLM are semantically rich, thereby alleviating the semantic over-simplification issue. Besides, to alleviate the low-density scene graph issue, we additionally incorporate a paraphrased version of the original caption. To this end, we further design a prompt for paraphrasing the original caption and extracting more triplets from the paraphrased caption. However, entities and predicates in the triplets obtained after Chain-1 are not yet aligned with the target data. Hence, we design a another prompt to align them with entity/predicate classes of interest, and ask the LLM to align them with semantically relevant lexeme included in a predefined lexicon, which is the set of vocabularies that are present in the target data (**Chain-2**). To further engage the LLM in the reasoning process of Chain-1 and Chain-2, we employ the in-context few-shot learning that incorporates a few input-output examples within the prompt, enabling the LLM to perform the task without the need for fine-tuning.

To validate the effectiveness of LLM4SGG, we apply it to state-of-the-art WSSGG methods (Zhang et al., 2023; Zhong et al., 2021). Through extensive experiments, we show that LLM4SGG significantly enhances the performance of existing WSSGG methods in terms of mean Recall@K and Recall@K performance on Visual Genome and GQA datasets by alleviating the semantic over-simplification and the low-density scene graph (See Table 1(c) where the number of triplets increased to 334K). A further appeal of LLM4SGG is that it is data-efficient, i.e., it outperforms state-of-the-art baselines even with a small amount of training images, verifying the effectiveness of LLM4SGG.

In summary, we make the following contributions:

- We identify two major issues overlooked by existing WSSGG studies, i.e., semantic over-simplification and low-density scene graph.

- We leverage an LLM along with the CoT strategy and the in-context few-shot learning technique to extract informative triplets without the need for fine-tuning the LLM. To the best of our knowledge, we are the first to leverage an LLM for the SGG task.

- LLM4SGG outperforms the state-of-the-art WSSGG methods, especially in terms of mR@K, demonstrating its efficacy in addressing the long-tail problem in WSSGG for the first time.

---

[2]We use the CoT strategy as a means to arrive at an answer in a stepwise manner, which differs from the Chain-of-Thought prompting.

## 2 RELATED WORK

**Weakly-Supervised Scene Graph Generation (WSSGG).** The WSSGG task aims to train an SGG model without relying on an annotated scene graph dataset. To achieve this, most WSSGG studies (Zhong et al., 2021; Ye & Kovashka, 2021; Li et al., 2022b; Zhang et al., 2023) utilize image captions and ground unlocalized triplets with image regions. Specifically, VSPNet (Zareian et al., 2020) proposes the iterative graph alignment algorithm to reflect the high-order relations between unlocalized triplets and image regions. SGNLS (Zhong et al., 2021) uses a pre-trained object detector (Ren et al., 2015) to ground the entities in unlocalized triplets, which share the same classes with the output of object detectors. In addition to the information derived from the object detector, Li et al. (2022b) employs a pre-trained vision-language model (Li et al., 2021a) to capture the semantic interactions among objects. $VS^3$ (Zhang et al., 2023) uses a grounding-based object detector (Li et al., 2022a), which calculates the similarity between the entity text in unlocalized triplet and image region, thereby grounding the unlocalized triplets. However, these methods overlook the triplet formation process that leads to the semantic over-simplification (Step 2) and the low-density scene graph (Step 3). In this regard, existing methods result in a sub-optimal performance even when unlocalized triplets are correctly grounded in Step 4.

**Large Language Model (LLM).** LLMs have demonstrated remarkable transferability to various downstream tasks such as symbolic reasoning, arithmetic, and common-sense reasoning (Brown et al., 2020; Touvron et al., 2023; Chowdhery et al., 2022). Specifically, GPT-3 (175B) (Brown et al., 2020) stands as a cornerstone to break the line of numerous language tasks. Inspired by GPT-3, PaLM (540B) (Chowdhery et al., 2022), LLaMA (65B) (Touvron et al., 2023), OPT (175B) (Zhang et al., 2022b), and LaMDA (137B) (Thoppilan et al., 2022) have been subsequently introduced. More recently, advanced GPT models (e.g., GPT-4 (OpenAI, 2023b), ChatGPT (OpenAI, 2023a)) fine-tuned with human feedback have gained prominence and widely applied for diverse applications, e.g., planner of tools (Lu et al., 2023; Gao et al., 2023), mobile task automation (Wen et al., 2023). In this work, we employ the power of LLM (i.e., ChatGPT) to alleviate the two issues, i.e., semantic over-simplification and low-density scene graph, in the context of the WSSGG task.

**In-Context Few-shot Learning.** In-context few-shot learning incorporates a few input-output examples related to a target task, conditioning the LLM on the context of examples. Specifically, GPT-3 (Brown et al., 2020) pioneered the concept of in-context learning to facilitate an LLM as a versatile model on diverse tasks. This breakthrough has proliferated a plethora of research to leverage the in-context few-shot learning for various tasks (Yao et al., 2022; Lu et al., 2023; Mishra et al., 2021; Zhang et al., 2022c). More precisely, Chameleon (Lu et al., 2023) integrates a few examples to enhance its understanding of tool planning task. Mishra et al. (2021) utilizes positive and negative examples related to questions for question generation tasks. ReAct (Yao et al., 2022) incorporates examples of reasoning with action for solving decision-making tasks. Inspired by recent in-context few-shot learning approaches, we provide a few examples to LLMs to help 1) understand the process of triplet extraction from a caption (i.e., Step 2), and 2) align the entity/predicate classes with the target data (i.e., Step 3) in the context of the WSSGG task.

## 3 METHOD

In this section, we describe LLM4SGG in detail. We would like to emphasize that LLM4SGG mainly focuses on the triplet formation process conducted in Step 2 (parsing) and Step 3 (aligning), while existing WSSGG approaches mainly focus on Step 4 (grounding). We start by presenting the problem formulation of WSSGG (Section 3.1), followed by the prompt configuration (Section 3.2). Next, we introduce how LLMs are adopted to address the two issues of conventional WSSGG approaches when parsing the image caption (Section 3.3) and aligning the triplets in captions with entity/predicate classes of interest (Section 3.4). Finally, we ground the unlocalized triplets by associating them with bounding boxes (i.e., image regions) and train the SGG model using the localized triplets (Section 3.5). The overall pipeline of LLM4SGG is shown in Figure 2.

### 3.1 PROBLEM FORMULATION

In the fully supervised SGG task, we aim to detect a scene graph $\mathbf{G}_f = \{\mathbf{s}_i, \mathbf{p}_i, \mathbf{o}_i\}_{i=1}^{N_f}$ that consists of triplets given an image $\mathcal{I}$, where $N_f$ is the number of triplets in the image. $\mathbf{s}_i$ and $\mathbf{o}_i$ denote the $i^{\text{th}}$ subject and the object, respectively, whose bounding boxes are $\mathbf{s}_{i,b}, \mathbf{o}_{i,b}$, and entity classes are $\mathbf{s}_{i,c}, \mathbf{o}_{i,c} \in \mathcal{C}_e$, where $\mathcal{C}_e$ is the set of predefined entity classes in the target data. $\mathbf{p}_i$ denotes the predicate between $\mathbf{s}_i$ and $\mathbf{o}_i$, and its class is $\mathbf{p}_{i,c} \in \mathcal{C}_p$, where $\mathcal{C}_p$ is the set of predefined predicate

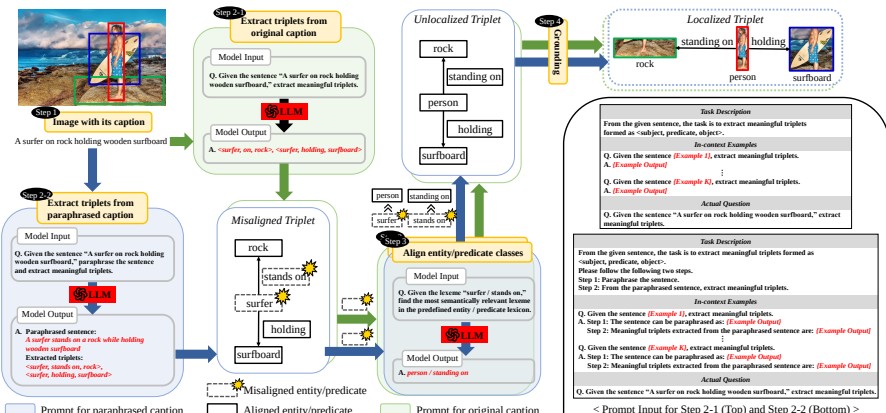

Figure 2: The pipeline of LLM4SGG. Given an image with its caption, we use an LLM to extract triplets from the original caption (Step 2-1) and the paraphrased caption (Step 2-2). Then, we align the entity/predicate classes within the extracted triplets with semantically similar lexeme in the target data via an LLM (Step 3), obtaining the unlocalized triplets. Lastly, we ground the unlocalized triplets over image regions (Step 4) followed by the training of an SGG model.

classes in the target data. By using the ground truth scene graphs as supervision, fully supervised SGG approaches train an SGG model $\mathcal{T}_\theta : \mathcal{I} \rightarrow \mathbf{G}_f$, which maps an image to a scene graph.

In the weakly supervised SGG task, we aim to generate a scene graph when the ground truth scene graph is not given, i.e., there are no bounding boxes and entity/predicate class information. Instead, existing WSSGG approaches (Zhang et al., 2023; Zhong et al., 2021; Li et al., 2022b; Ye & Kovashka, 2021; Shi et al., 2021) use image captions along with associated images to produce scene graphs, i.e., localized triplets. More precisely, they extract a set of triplets $\mathbf{G}_w = \{\mathbf{s}_i, \mathbf{p}_i, \mathbf{o}_i\}_{i=1}^{N_w}$ from the image captions, where $N_w$ is the number of triplets extracted from the captions. However, while the extracted triplets contain the class information (i.e., $\mathbf{s}_{i,c}$, $\mathbf{o}_{i,c}$ and $\mathbf{p}_{i,c}$), they are unlocalized, since bounding boxes $\mathbf{s}_{i,b}$ and $\mathbf{o}_{i,b}$ are not included in the caption. Therefore, it is essential to perform the grounding step to associate the unlocalized triplets with bounding boxes. Once we have obtained the localized triplets, we can apply the conventional SGG training scheme, i.e., $\mathcal{T}_\theta : \mathcal{I} \rightarrow \mathbf{G}_f$.

In this paper, our focus is to address the semantic over-simplification and low-density scene graph issues regarding the unlocalized triplets $\mathbf{G}_w$, which has been overlooked in existing WSSGG studies. Specifically, we aim to produce an enhanced $\mathbf{G}_w$ by refining the process of scene graph dataset construction via an LLM. This refinement includes the triplet extraction step from the caption (Step 2) and the alignment of entity/predicate classes (Step 3), leveraging the LLM's comprehensive understanding of language and reasoning ability.

## 3.2 PROMPT CONFIGURATION

In fact, it is not a trivial task for an LLM to immediately generate triplets from a caption whose entities and predicates are aligned with entity/predicate classes of interest, as such a task is a novel task for the LLM. Inspired by the idea of the Chain-of-thought (CoT) (Wei et al., 2022b), which arrives at an answer in a stepwise manner, we separate the triplet formation process into the following two chains: Chain-1 – Extracting triplets from captions. Chain-2 – Aligning entities and predicates with the entity/predicate classes of interest. To carefully design each chain, we define the LLM function, i.e., $LLM(\cdot)$, with the following prompt input:

$$Output = LLM(\underbrace{\text{Task description}, \text{In-context examples}, \text{Actual question}}_{\text{Prompt input}}), \quad (1)$$

where $LLM(\cdot)$ is the decoder of the LLM, generating the $Output$ given the prompt input. The prompt input consists of three components in a sequence: 1) task description, i.e., the delineation of the task that we intend to perform, 2) in-context examples, i.e., sample questions and answers related to the task at hand, 3) actual question, i.e., an inquiry from which we intend to derive the answer. Note that *in-context examples* is closely related to the in-context few-shot learning (Wei et al., 2022b; Brown et al., 2020; Wei et al., 2022a), which is shown to enhance the LLM's understanding of the task. Note that the above configuration of the prompt input is applied to the triplet extraction (Chain-1) (Section 3.3) and the alignment of entity/predicate classes (Chain-2) (Section 3.4).

### 3.3 CHAIN-1. TRIPLET EXTRACTION VIA LLM (STEP 2 IN FIGURE 2)

Based on the LLM's comprehensive understanding of the context of an image caption, we aim to extract triplets from the caption. As discussed in Section 1, we use not only the original caption but also a paraphrased caption generated by the LLM to address the low-density scene graph issue.

**Extracting triplets from a *paraphrased* caption (Step 2-2).** To extract triplets from a paraphrased caption, we inform the LLM about the task at hand by presenting the following prompt: FROM THE GIVEN SENTENCE, THE TASK IS TO EXTRACT MEANINGFUL TRIPLETS FORMED AS ⟨SUBJECT, PREDICATE, OBJECT⟩ (i.e., Task description in Equation 1). We then instruct the LLM to follow the two steps, i.e., paraphrasing step and triplet extraction step. To help the LLM understand the process of performing the two steps, we present few examples to the LLM that describe how to answer the questions (i.e., In-context examples in Equation 1[3]), which involves a manual construction of questions and corresponding answers to the paraphrasing and the triplet extraction steps. That is, for given a caption, we show the LLM how we expect a paraphrased caption and the extracted triplets would look like (e.g., Given "Four clocks sitting on a floor next to a woman's feet," we show the LLM that a paraphrased sentence would be "Four clocks are placed on the floor beside a woman's feet," and extracted triplets would be ⟨clocks, placed on, floor⟩ and ⟨clocks, beside, feet⟩). Lastly, we show the caption of our interest to the LLM, and let the LLM extract triplets from the caption (i.e., Actual question in Equation 1). Please refer to Figure 2 right (bottom) for an example of the prompt input used in Step 2-2.

**Extracting triplets from the *original* caption (Step 2-1).** As extracting triplets from the original caption does not involve the caption paraphrasing step, we exclude it from the prompt used in Step 2-2. Please refer to Figure 2 right (top) for an example of the prompt input used in Step 2-1.

In summary, we obtain triplets from both the original and paraphrased captions after Step 2-1 and Step 2-2, respectively, which in turn alleviates the semantic over-simplification issue of predicates and the low-density scene graph issue.

### 3.4 CHAIN-2. ALIGNMENT OF CLASSES IN TRIPLETS VIA LLM (STEP 3 IN FIGURE 2)

The entities (i.e., subject and object) and predicates within the triplets obtained from Step 2 described in Section 3.3 are not yet aligned with the target data. Based on the semantic reasoning ability of the LLM, we aim to align them with the semantically relevant lexeme in the target data.

**Aligning entities in the triplets with entity classes of interest.** We instruct the LLM with the following prompt: GIVEN THE LEXEME {ENTITY}, FIND SEMANTICALLY RELEVANT LEXEME IN THE PREDEFINED ENTITY LEXICON, where the predefined entity lexicon is $\mathcal{C}_e$ (i.e., Task description in Equation 1). Similar to Section 3.3, we present a few examples to the LLM that describe how to answer the questions (i.e., In-context examples in Equation 1). For example, we provide the LLM with a few examples regarding hierarchical relationships such as pigeon being semantically relevant to bird, and singular-plural relationships such as surfboards being semantically relevant to surfboard. Lastly, we show the entity of our interest to the LLM (i.e., Actual question in Equation 1), which enables the LLM to generate an answer by finding the most semantically relevant entity in $\mathcal{C}_e$. Please refer to Table 7 in Appendix A.1 for an example of the prompt input.

**Aligning predicates in the triplets with predicate classes of interest.** Likewise, we instruct the LLM with the following prompt: GIVEN THE LEXEME {PREDICATE}, FIND SEMANTICALLY RELEVANT LEXEME IN THE PREDEFINED PREDICATE LEXICON, where the predefined predicate lexicon is $\mathcal{C}_p$ (i.e., Task description in Equation 1). We also present a few examples to the LLM that describe how to answer the questions (i.e., In-context examples in Equation 1). For example, we provide the LLM with a few examples regarding tense relationships such as the lies on being semantically relevant to lying on, and positional relationship such as next to being semantically relevant to near. Lastly, we show the predicate of our interest to the LLM (i.e., Actual question in Equation 1). Please refer to Table 8 in Appendix A.1 for an example of the prompt input.

After performing Step 2 (Section 3.3) and Step 3 (Section 3.4), we obtain intermediate unlocalized triplets $\hat{\mathbf{G}}_w = \{\mathbf{s}_i, \mathbf{p}_i, \mathbf{o}_i\}_{i=1}^{\hat{N}_w}$, where $\mathbf{s}_{i,c}, \mathbf{o}_{i,c} \in \{\mathcal{C}_e \cup \text{None}\}$ and $\mathbf{p}_{i,c} \in \{\mathcal{C}_p \cup \text{None}\}$. It is worth noting that if there is no semantically relevant lexeme, we request the LLM to generate None as the answer, due to the fact that the entity/predicate classes in the target data may not cover a wide range of entities/predicates. Similar to the conventional approach, we discard a triplet if any of its three

---

[3]We use captions in COCO caption dataset for examples.

components (i.e., subject, predicate and object) is None. Lastly, we obtain the final unlocalized triplets $\mathbf{G}_w = \{\mathbf{s}_i, \mathbf{p}_i, \mathbf{o}_i\}_{i=1}^{N_w}$ ($N_w \leq \hat{N}_w$), where $\mathbf{s}_{i,c}, \mathbf{o}_{i,c} \in \mathcal{C}_e$ and $\mathbf{p}_{i,c} \in \mathcal{C}_p$.

## 3.5 MODEL TRAINING

Given the final unlocalized triplets $\mathbf{G}_w$, we ground them over relevant image regions to get localized triplets, meaning that we obtain $\mathbf{s}_{i,b}$ and $\mathbf{o}_{i,b}$. To this end, we employ two state-of-the-art grounding methods, i.e., SGNLS (Zhong et al., 2021) and VS$^3$ (Zhang et al., 2023). Please refer to Appendix A.2 for more detail about how each method performs grounding. After grounding $\mathbf{G}_w$, we obtain localized triplets and use them as pseudo-labels for training a supervised SGG model. Please refer to Appendix A.3 for more detail about the model training.

## 4 EXPERIMENT

**Datasets.** To **train** an SGG model without an annotated scene graph dataset, we use the COCO caption dataset (Chen et al., 2015), which is commonly used for WSSGG studies (Zhang et al., 2023; Li et al., 2022b; Zhong et al., 2021; Ye & Kovashka, 2021). For fair comparisons, we use the same set of 64K images that have been utilized in previous WSSGG studies (Li et al., 2022b; Zhong et al., 2021). Each image is associated with five captions. To **evaluate** the trained SGG model, we employ the widely used Visual Genome (VG) dataset (Krishna et al., 2017) and GQA dataset (Hudson & Manning, 2019). The VG dataset contains the ground-truth localized triplet information annotated by humans. We follow the standard split of VG (Xu et al., 2017), which consist of 150 entity classes and 50 predicate classes. For the GQA dataset used in the SGG task, we follow the same pre-processing step of a previous SGG study (Dong et al., 2022), which involves selecting top-200 frequent entity classes and top-100 frequent predicate classes. In both datasets, 30% of the total images are used for evaluation. Please refer to Appendix C.1 for more details regarding the datasets.

**Evaluation metrics.** Recent fully-supervised SGG studies (Yoon et al., 2023; Zhang et al., 2022a; Biswas & Ji, 2023) have emphasized improving the accuracy of predictions for fine-grained predicates rather than coarse-grained predicates, since the former construct richer scene graphs. As a result, they commonly use mean Recall@K (mR@K) that computes the average of Recall@K (R@K) across all predicates. In line with the recent emphasis on fine-grained predicates, we incorporate both mR@K and R@K in our evaluation, whereas previous WSSGG studies (Zhang et al., 2023; Li et al., 2022b; Ye & Kovashka, 2021) mainly rely on the R@K metric alone. Moreover, we report F@K, which is the harmonic average of R@K and mR@K to jointly consider R@K and mR@K, following previous SGG studies (Khandelwal & Sigal, 2022; Zhang et al., 2022a). Regarding the evaluation task, we follow previous WSSGG studies and adopt the Scene Graph Detection (SGDet) task, where both the ground-truth bounding box and the entity class information are not provided. Please refer to Appendix C.2 for more detail regarding the task.

**Baselines.** Please refer to Appendix C.3 for details regarding the baselines.

**Implementation Details.** In the grounding process in SGNLS (Zhong et al., 2021), we used Faster R-CNN (Ren et al., 2015) as the object detector, which is pre-trained on Open Images (Kuznetsova et al., 2020). In the grounding process in VS$^3$, we used GLIP (Li et al., 2022a) as the object detector with the Swin-L backbone (Liu et al., 2021). Regarding an LLM, we use *gpt-3.5-turbo* in ChatGPT (OpenAI, 2023a). Note that to further alleviate the long-tailed predicate distribution after Step 3 in our framework, we select the most fine-grained predicate when there are multiple predicates between the same subject-object pair, where the fine-grainedness is determined based on the predicate distribution within the entire set of unlocalized triplets. For more insights regarding the impact of the predicate selection, please refer to Appendix D.2.

## 4.1 QUANTITATIVE RESULT ON VG

Table 2 shows the performance of baseline models and those when LLM4SGG is applied. We have the following observations: **1)** Applying LLM4SGG to SGNLS and VS$^3$ improves the performance in terms of R@K and mR@K, which demonstrates the effectiveness of the triplet formation through LLM4SGG. Notably, LLM4SGG significantly improves mR@K, implying that LLM4SGG effectively alleviates the long-tailed problem in WSSGG.

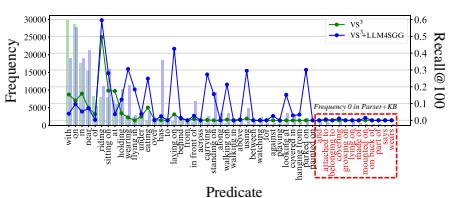

Figure 3: Per class performance (Bar: num. predicate instances, Line: Recall@100).

Table 2: Performance comparisons on the SGDet task. The best performance among WSSGG models is in bold. The red numbers indicate the absolute performance improvement after applying LLM4SGG. *Rwt* denotes using the reweighting strategy (Zhang et al., 2022a).

| Method | R@50 | R@100 | mR@50 | mR@100 | F@50 | F@100 |
|---|---|---|---|---|---|---|
| Motif *(CVPR'18)* - Fully-supervised | 31.89 | 36.36 | 6.38 | 7.57 | 10.63 / 12.53 | 12.53 |
| LSWS *(CVPR'21)* | 3.29 | 3.69 | 3.27 | 3.66 | 3.28 | 3.67 |
| SGNLS *(ICCV'21)* | 3.80 | 4.46 | 2.51 | 2.78 | 3.02 | 3.43 |
| SGNLS *(ICCV'21)*+LLM4SGG | $5.09_{+1.29}$ | $5.97_{+1.51}$ | $4.08_{+1.57}$ | $4.49_{+1.71}$ | $4.53_{+1.51}$ | $5.13_{+1.70}$ |
| Li et al *(MM'22)* | 6.40 | 7.33 | 1.73 | 1.98 | 2.72 | 3.12 |
| VS$^3$ *(CVPR'23)* | 6.60 | 8.01 | 2.88 | 3.25 | 4.01 | 4.62 |
| VS$^3$ *(CVPR'23)*+LLM4SGG | $\mathbf{8.91}_{+2.31}$ | $\mathbf{10.43}_{+2.42}$ | $7.11_{+4.23}$ | $8.18_{+4.93}$ | $\mathbf{7.91}_{+3.90}$ | $\mathbf{9.17}_{+4.55}$ |
| VS$^3$ *(CVPR'23)*+Rwt | 4.25 | 5.04 | 5.17 | 5.99 | 4.67 | 5.47 |
| VS$^3$ *(CVPR'23)*+Rwt+LLM4SGG | $5.10_{+0.85}$ | $6.34_{+1.30}$ | $\mathbf{8.42}_{+3.25}$ | $\mathbf{9.90}_{+3.91}$ | $6.35_{+1.69}$ | $7.73_{+2.26}$ |

This can be clearly seen in Figure 3, which shows the performance gain on fine-grained predicates. **2)** VS$^3$+Rwt+LLM4SGG further improves mR@K of VS$^3$+Rwt. We attribute this to the fact that the conventional approach generates a limited number of fine-grained predicates, which makes the reweighting strategy less effective within VS$^3$. Especially, non-existent predicates can never be predicted even when the reweighting strategy is applied. On the other hand, LLM4SGG increases the number of instances that belong to fine-grained predicates, which is advantageous for the reweighting strategy. For the per class performance comparison over the reweighting stregty, please refer to Appendix D.1. **3)** The performance gain obtained from applying LLM4SGG is greater on VS$^3$ (i.e., VS$^3$+LLM4SGG) than on SGNLS (i.e., SGNLS+LLM4SGG). The major reason lies in the difference in how SGNLS and VS$^3$ make use of the pool of 344K unlocalized triplets obtained through LLM4SGG. Specifically, in the grounding process of SGNLS, we observe that 100K out of 344K unlocalized triplets (i.e., 29%) fail to be grounded, and thus not used for training. On the other hand, VS$^3$ successfully grounds all 344K unlocalized triplets and fully utilize them for training, allowing it to fully enjoy the effectiveness of LLM4SGG. This indicates that LLM4SGG makes synergy when paired with a grounding method that is capable of fully utilizing the unlocalized triplets. For more details regarding the impact of grounding methods, please refer to Appendix B.

## 4.2 ABLATION STUDIES

In Table 3, we conduct ablation studies on VG dataset to understand the effectiveness of each component of LLM4SGG, where VS$^3$ is used as the grounding method. Note that row (a) is equivalent to vanilla VS$^3$. We have the following observations. **1) Effect of using the paraphrased caption:** Including the paraphrased caption in addition to the original caption (row (b)) increases the number of triplets

Table 3: Ablation studies. (PC: Using Paraphrased Caption in addition to the original caption / LP: LLM-based Parsing / LA: LLM-based Alignment)

| Row | PC | LP | LA | # Triplet | R@50 / 100 | mR@50 / 100 | F@50 / 100 |
|---|---|---|---|---|---|---|---|
| (a) | | | | 154K | 6.60 / 8.01 | 2.88 / 3.25 | 4.01 / 4.62 |
| (b) | ✓ | | | 243K | 9.46 / 11.22 | 3.43 / 3.92 | 5.03 / 5.81 |
| (c) | ✓ | ✓ | | 256K | 8.42 / 9.85 | 5.99 / 6.95 | 7.00 / 8.15 |
| (d) | ✓ | | ✓ | 327K | **11.76 / 13.38** | 3.50 / 4.05 | 5.39 / 6.22 |
| (e) | ✓ | ✓ | ✓ | 344K | 8.91 / 10.43 | **7.11 / 8.18** | **7.91 / 9.17** |

(154K→243K), resulting in an improved overall performance. This demonstrates that the paraphrased caption alleviates the low-density scene graph issue. **2) Effect of LLM-based parsing:** The LLM-based parsing (row (c)) for extracting triplets improves mR@K of row (b). This indicates that the LLM-based parsing increases the number of instances that belong to fine-grained predicates, which in turn alleviates the semantic over-simplification issue. **3) Effect of LLM-based alignment:** The LLM-based alignment (row (d)) of entities/predicates in the extracted triplets increases the number of triplets from 243K to 327K (row (b) vs (d)), which indicates that the low-density scene graph issue is alleviated. Consequently, R@K and mR@K of row (d) are greater than those of row (b). **4)** The fully-fledged approach (row (e)) generally improves R@K and mR@K, showing the best performance in terms of F@K. It is important to highlight that when using the LLM-based parsing, the performance of mR@K significantly increases with a moderate decrease in R@K. This trade-off is attributed to the fact that R@K generally improves when the coarse-grained predicates are dominant (Zhang et al., 2022a). In contrast, our approach, which addresses the semantic over-simplification issue, decreases the number instances that belong to coarse-grained predicates while simultaneously increasing those that belong to fine-grained predicates (Figure 1(b)), which in turn results in a substantial improvement in mR@K. We would like to emphasize that the performance in terms of mR@K is crucial in the context of SGG research (Yoon et al., 2023; Biswas & Ji, 2023; Dong et al., 2022), as fine-grained predicates offer richer information.

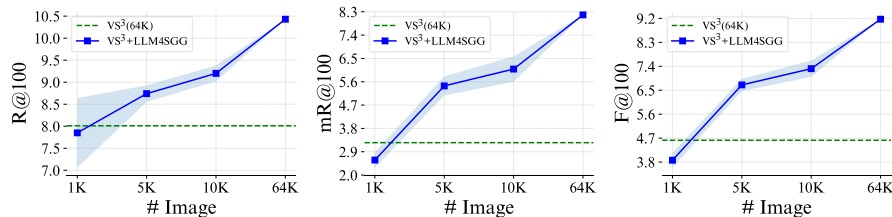

Figure 4: Performance over various numbers of images used for training VS$^3$+LLM4SGG.

### 4.3 ANALYSIS OF DATA-EFFICIENCY

To assess the effectiveness of LLM4SGG under the lack of available training data, we conduct experiments given a limited number of images. Specifically, among 64K images used for training VS$^3$ and VS$^3$+LLM4SGG in Table 2, we randomly sample 1K (1.5%), 5K (7.8%), and 10K (15.6%) images with replacement, and train VS$^3$+LLM4SGG for five times. Figure 4 shows the average performance over various numbers of images used for training VS$^3$+LLM4SGG along with the variance (in blue area). We observe that when using only 1K images, the performance is slightly inferior to the baseline that used 64K for training (i.e., VS$^3$). However, as we increase the number of images used for training to 5K images, we observe a significant improvement in both R@K and mR@K compared with the baseline. This demonstrates that LLM4SGG is data-efficient, as *it outperforms the baseline even with only 7.8% of the total images used for training the baseline.* Moreover, when further increasing the number of images to 10K, we observe further performance improvements, and when it reaches 64K, which is the same as the number of images used for training the baseline, the performance is the best. In summary, LLM4SGG enables data-efficient model training even with a limited amount of available images for training, thanks to alleviating the semantic over-simplification and low-density scene graph issues.

### 4.4 QUANTITATIVE RESULT ON GQA

In Table 4, we additionally conducted experiments on GQA dataset (Hudson & Manning, 2019). Please refer to Appendix E.1 for more detailed descriptions on the training and evaluation processes on GQA dataset. The GQA dataset contains twice as many predicates as the Visual

Table 4: Performance comparison on GQA.

| Method | R@50 / 100 | mR@50 / 100 | F@50 / 100 |
|---|---|---|---|
| Motif (Fully-supervised) | 28.90 / 33.10 | 6.40 / 7.70 | 10.48 / 12.49 |
| VS$^3$ | 5.90 / 6.97 | 1.60 / 1.81 | 2.52 / 2.87 |
| VS$^3$+LLM4SGG | **8.88 / 10.38** | **5.33 / 6.51** | **6.66 / 8.00** |

Genome dataset and includes complicated predicates (e.g., sitting next to, standing in front of). As a result, when obtaining unlocalized triplets using the conventional WSSGG approach, we observe that 44 out of 100 predicates have a frequency of 0, and the predicate distribution is extremely long-tailed. Consequently, the baseline (i.e., VS$^3$) exhibits significantly lower performance, especially in terms of mR@K. On the other hand, our approach shows substantial performance improvements not only in R@K but also in mR@K, thanks to the mitigation of semantic over-simplification and low-density scene graph issues. Please refer to Appendix E.2 for the predicate distribution and performance comparison for each class in GQA dataset. Additionally, please refer to Appendix E.3 for qualitative analyses on GQA dataset.

## 5 CONCLUSION & FUTURE WORK

In this work, we focus on the triplet formation process in WSSGG, whose importance is overlooked by previous studies. To alleviate the semantic over-simplification and low-density scene graph issues inherent in the triplet formation process, we propose a new approach, i.e., LLM4SGG, which leverages a pre-trained LLM during the extraction of triplets from the captions, and alignment of entity/predicate classes with those in the target data. It is important to note that construction of these triplets is not required every time to train the SGG model; instead, it is a one-time pre-processing step. In this regard, we contribute to generating enhanced triplets compared to the conventional approach. As a result, we outperform baselines in terms of R@K, mR@K and F@K on Visual Genome and GQA datasets. For future work, an LLM can be used to ground the unlocalized triplets in Step 4. Recently, vision-language representation learning has been developed for transforming visual features into textual features to facilitate the use of visual features as input to an LLM (Li et al., 2023a; Zhu et al., 2023). In this regard, given the visual features of bounding boxes as input, we could ask the LLM to identify relevant bounding boxes based on the textual information of entities within unlocalized triplets using the comprehensive understanding of the context of triplets.

ETHICS STATEMENT

Regarding the adherence of ICLR Code of Ethics, to the best of our knowledge, we have no violated ethical issues throughout this paper. All datasets and pre-trained models used for experiments are publicly available.

REPRODUCIBILITY STATEMENT

Our experiments incorporate the utilization of an LLM, i.e., ChatGPT. To further improve reproducibility of experiment results, we set temperature to its minimum value of 0 in ChatGPT, which controls the variance of text generation. Additionally, we provide full prompts used for triplet extraction process in Appendix A.1, and the source code in https://anonymous.4open.science/r/LLM4SGG-83DD along with the accessible datasets, pre-processed triplets constructed by LLM4SGG, and our pre-trained model.

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

# *Supplementary Material*

# A  DETAILS OF METHOD

## A.1  DETAILS OF PROMPT

We provide the complete prompts for extracting triplets from paraphrased captions (Table 5) and from original captions (Table 6). Moreover, we provide those for aligning entity classes (Table 7), and predicate classes (Table 8) with entity and predicate classes in the target data, respectively.

Table 5: Prompt for triplet extraction from paraphrased caption.

| **Task Description** |
|---|
| From the given sentence, the task is to extract meaningful triplets formed as <subject, predicate, object>. |
| To extract meaningful triplets from the sentence, please follow the following two steps. |
| **Step 1**: Paraphrase the sentence. |
| **Step 2**: From the paraphrased sentence obtained in the Step 1, extract meaningful triplets formed as <subject, predicate, object>. |
| Note that the subject is the entity or noun that performs the action or is being described, and the object is the entity or noun that is |
| affected by the action or is receiving the action. The predicate is a verb or adjective without auxiliary verb. |
| **In-context Examples** |
| Let's take a few examples to understand how to extract meaningful triplets. |
| Question: Given the sentence "a slice of bread is covered with a sour cream and guacamole," extract meaningful triplets. |
| Answer:  Step 1: The sentence can be paraphrased as: |
| A piece of bread is topped with both sour cream and guacamole. |
| Step 2: Meaningful triplets, where the subject and object are the simple noun, extracted from the paraphrased sentence are: |
| <bread, topped with, sour cream>, <bread, topped with, guacamole>. |
| The meaningful triplets are *<bread, topped with, sour cream>, and <bread, topped with, guacamole>*. |
| Question: Given the sentence "A beautiful woman walking a dog on top of a beach," extract meaningful triplets. |
| Answer:  Step 1: The sentence can be paraphrased as: |
| A lovely woman strolling with a dog on the beach. |
| Step 2: Meaningful triplets, where the subject and object are the simple noun, extracted from the paraphrased sentence are: |
| <woman, strolling with, dog>, <woman, on, beach>, <dog, on, beach>. |
| The meaningful triplets are *<woman, strolling with, dog>, <woman, on, beach>, and <dog, on, beach>*. |
| Question: Given the sentence "Four clock sitting on a floor next to a woman's feet," extract meaningful triplets. |
| Answer:  Step 1: The sentence can be paraphrased as: |
| Four clocks are placed on the floor beside a woman's feet. |
| Step 2: Meaningful triplets, where the subject and object are the simple noun, extracted from the paraphrased sentence are: |
| <clocks, placed on, floor>, <clocks, beside, feet>. |
| The meaningful triplets are *<clocks, placed on, floor> and <clocks, beside, feet>*. |
| Question: Given the sentence "One person sits in a chair looking at her phone while another rests on the couch," extract meaningful triplets. |
| Answer:  Step 1: The sentence can be paraphrased as: |
| A person is seated in a chair, using their phone, while someone else is relaxing on the couch. |
| Step 2: Meaningful triplets, where the subject and object are the simple noun, extracted from the paraphrased sentence are: |
| <person, seated in, chair>, <person, using, phone>, <person, relaxing on, couch>. |
| The meaningful triplets are *<person, seated in, chair>, <person, using, phone>, and <person, relaxing on, couch>*. |
| Question: Given the sentence "A lady and a child near a park bench with kites and ducks flying in the sky and on the ground," extract meaningful triplets. |
| Answer:  Step 1: The sentence can be paraphrased as: |
| A woman and a child are close to a park bench, while kites soar through the sky and ducks move around on the ground. |
| Step 2: Meaningful triplets, where the subject and object are the simple noun, extracted from the paraphrased sentence are: |
| <woman, close to, park bench>, <child, close to, park bench>, <kites, soar through, sky>, <ducks, move around, ground>. |
| The meaningful triplets are *<woman, close to, park bench>, <child, close to, park bench>, <kites, soar through, sky>, and <ducks, move around, ground>*. |
| Question: Given the sentence "Two men sit on a bench near the sidewalk and one of them talks on a cell phone," extract meaningful triplets. |
| Answer:  Step 1: The sentence can be paraphrased as: |
| Two guys are seated on a bench near the road, and one of them talks on a mobile phone. |
| Step 2: Meaningful triplets, where the subject and object are the simple noun, extracted from the paraphrased sentence are: |
| <guys, seated on, bench>, <bench, near, road>, <guy, talks on, phone>. |
| The meaningful triplets are *<guys, seated on, bench>, <bench, near, road>, and <guy, talks on, phone>*. |
| **Actual Question** |
| Question: Given the sentence Input, extract meaningful triplets. Answer: |

Table 6: Prompt for triplet extraction from original caption.

| **Task Description** |
|---|
| From the given sentence, the task is to extract meaningful triplets formed as <subject, predicate, object>. |
| Note that the subject is the entity or noun that performs the action or is being described, and the object is the entity or noun that is affected by the action |
| or is receiving the action. The predicate is a verb or adjective without auxiliary verb, and is represented without the tense. |
| **In-context Examples** |
| Let's take a few examples to understand how to extract meaningful triplets. |
| Question: Given the sentence "a slice of bread is covered with a sour cream and guacamole," extract the meaningful triplets. |
| Answer: Meaningful triplets are *<bread, covered with, sour cream>, and <bread, covered with, guacamole>*. |
| Question: Given the sentence "A beautiful woman walking a dog on top of a beach," extract meaningful triplets. |
| Answer: Meaningful triplets are *<woman, walking with, dog>, <woman, on, beach>, and <dog, on, beach>*. |
| Question: Given the sentence "Four clock sitting on a floor next to a woman's feet," extract the meaningful triplets. |
| Answer: Meaningful triplets are *<clock, sitting on, floor> and <clock, next to, feet>*. |
| Question: Given the sentence "One person sits in a chair looking at her phone while another rests on the couch," extract meaningful triplets. |
| Answer: Meaningful triplets are *<person, sits in, chair>, <person, looking at, phone>, and <person, rests on, couch>*. |
| Question: Given the sentence "A lady and a child near a park bench with kites and ducks flying in the sky and on the ground," extract meaningful triplets. |
| Answer:  Meaningful triplets are *<lady, near, park bench>, <child, near, park bench>, <kites, flying in sky>, and <ducks, on, ground>*. |
| Question: Given the sentence "Two men sit on a bench near the sidewalk and one of them talks on a cell phone," extract meaningful triplets. |
| Answer:  Meaningful triplets are *<men, sit on, bench>, <bench, near, sidewalk>, and <man, talks on, phone>*. |
| **Actual Question** |
| Question: Given the sentence Input, extract meaningful triplets. Answer: |

## A.2  DETAILS OF GROUNDING METHODS

To ground unlocalized triplets, we employ two state-of-the-art grounding methods, i.e., SGNLS (Zhong et al., 2021) and VS[3] (Zhang et al., 2023). Herein, we provide a detailed explanation of each grounding method.

Table 7: Prompt for alignment of entity class.

| Task Description |
|---|
| The predefined entity lexicon containing 150 lexemes is numbered as follows: 1.airplane 2.animal 3.arm 4.bag 5.banana 6.basket 7.beach 8.bear 9.bed 10.bench 11.bike 12.bird 13.board 14.boat 15.book 16.boot 17.bottle 18.bowl 19.box 20.boy 21.branch 22.building 23.bus 24.cabinet 25.cap 26.car 27.cat 28.chair 29.child 30.clock 31.coat 32.counter 33.cow 34.cup 35.curtain 36.desk 37.dog 38.door 39.drawer 40.ear 41.elephant 42.engine 43.eye 44.face 45.fence 46.finger 47.flag 48.flower 49.food 50.fork 51.fruit 52.giraffe 53.girl 54.glass 55.glove 56.guy 57.hair 58.hand 59.handle 60.hat 61.head 62.helmet 63.hill 64.horse 65.house 66.jacket 67.jean 68.kid 69.kite 70.lady 71.lamp 72.laptop 73.leaf 74.leg 75.letter 76.light 77.logo 78.man 79.men 80.motorcycle 81.mountain 82.mouth 83.neck 84.nose 85.number 86.orange 87.pant 88.paper 89.paw 90.people 91.person 92.phone 93.pillow 94.pizza 95.plane 96.plant 97.plate 98.player 99.pole 100.post 101.pot 102.racket 103.railing 104.rock 105.roof 106.room 107.screen 108.seat 109.sheep 110.shelf 111.shirt 112.shoe 113.short 114.sidewalk 115.sign 116.sink 117.skateboard 118.ski 119.skier 120.sneaker 121.snow 122.sock 123.stand 124.street 125.surfboard 126.table 127.tail 128.tie 129.tile 130.tire 131.toilet 132.towel 133.tower 134.track 135.train 136.tree 137.truck 138.trunk 139.umbrella 140.vase 141.vegetable 142.vehicle 143.wave 144.wheel 145.window 146.windshield 147.wing 148.wire 149.woman 150.zebra. 
 Given the lexeme, the task is to find semantically relevant lexeme from the predefined entity lexicon. 
 However, if there is no semantically relevant lexeme in the predefined entity lexicon, please answer 0.None. |
| **In-context Examples** |
| Let's take a few examples. 
 Question: Given the lexeme "water," find semantically relevant lexeme in the predefined entity lexicon. Answer: *0.None* 
 Question: Given the lexeme "bus," find semantically relevant lexeme in the predefined entity lexicon. Answer: *142.vehicle* 
 Question: Given the lexeme "steel," find semantically relevant lexeme in the predefined entity lexicon. Answer: *0.None* 
 Question: Given the lexeme "vanity," find semantically relevant lexeme in the predefined entity lexicon. redAnswer: *110.shelf* 
 Question: Given the lexeme "desktop," find semantically relevant lexeme in the predefined entity lexicon. Answer: *72.laptop* 
 Question: Given the lexeme "cobble," find semantically relevant lexeme in the predefined entity lexicon. Answer: *104.rock* 
 Question: Given the lexeme "poles," find semantically relevant lexeme in the predefined entity lexicon. Answer: *99.pole* 
 Question: Given the lexeme "wastebasket," find semantically relevant lexeme in the predefined entity lexicon. Answer: *6.basket* 
 Question: Given the lexeme "blue," find semantically relevant lexeme in the predefined entity lexicon. Answer: *0.None* 
 Question: Given the lexeme "motorcyclist," find semantically relevant lexeme in the predefined entity lexicon. Answer: *98.player* 
 Question: Given the lexeme "passenger," find semantically relevant lexeme in the predefined entity lexicon. Answer: *91.person* 
 Question: Given the lexeme "pigeon," find semantically relevant lexeme in the predefined entity lexicon. Answer: *12.bird* 
 Question: Given the lexeme "grass," find semantically relevant lexeme in the predefined entity lexicon. Answer: *96.plant* 
 Question: Given the lexeme "surfboards," find semantically relevant lexeme in the predefined entity lexicon. Answer: *125.surfboard* 
 Question: Given the lexeme "striped shirts," find semantically relevant lexeme in the predefined entity lexicon. Answer: *111.shirt* |
| **Actual Question** |
| Question: Given the lexeme Input, find semantically relevant lexeme in the predefined entity lexicon. Answer: |

Table 8: Prompt for alignment of predicate class.

| Task Description |
|---|
| The predefined predicate lexicon containing 50 lexemes is numbered as follows: 1.above 2.across 3.against 4.along 5.and 6.at 7.attached to 8.behind 9.belonging to 10.between 11.carrying 12.covered in 13.covering 14.eating 15.flying in 16.for 17.from 18.growing on 19.hanging from 20.has 21.holding 22.in 23.in front of 24.laying on 25.looking at 26.lying on 27.made of 28.mounted on 29.near 30.of 31.on 32.on back of 33.over 34.painted on 35.parked on 36.part of 37.playing 38.riding 39.says 40.sitting on 41.standing on 42.to 43.under 44.using 45.walking in 46.walking on 47.watching 48.wearing 49.wears 50.with. 
 Given the lexeme, the task is to find semantically relevant lexeme from the predefined predicate lexicon. 
 However, if there is no semantically relevant lexeme in the predefined predicate lexicon, please answer 0.None. |
| **In-context Examples** |
| Let's take a few examples. 
 Question: Given the lexeme "next to," find semantically relevant lexeme in the predefined predicate lexicon. Answer: *29.near* 
 Question: Given the lexeme "are parked in," find semantically relevant lexeme in the predefined predicate lexicon. Answer: *35.parked on* 
 Question: Given the lexeme "waiting," find semantically relevant lexeme in the predefined predicate lexicon. Answer: *0.None* 
 Question: Given the lexeme "sitting," find semantically relevant lexeme in the predefined predicate lexicon. redAnswer: *40.sitting on* 
 Question: Given the lexeme "grazing," find semantically relevant lexeme in the predefined predicate lexicon. Answer: *14.eating* 
 Question: Given the lexeme "pointing to," find semantically relevant lexeme in the predefined predicate lexicon. Answer: *0.None* 
 Question: Given the lexeme "lies on," find semantically relevant lexeme in the predefined predicate lexicon. Answer: *24.lying on* 
 Question: Given the lexeme "sitting underneath," find semantically relevant lexeme in the predefined predicate lexicon. Answer: *43.under* 
 Question: Given the lexeme "placed next to," find semantically relevant lexeme in the predefined predicate lexicon. Answer: *29.near* 
 Question: Given the lexeme "looking down at," find semantically relevant lexeme in the predefined predicate lexicon. Answer: *25.looking at* 
 Question: Given the lexeme "containing," find semantically relevant lexeme in the predefined predicate lexicon. Answer: *0.has* 
 Question: Given the lexeme "perched on," find semantically relevant lexeme in the predefined predicate lexicon. Answer: *40.sitting on* 
 Question: Given the lexeme "driving," find semantically relevant lexeme in the predefined predicate lexicon. Answer: *0.None* 
 Question: Given the lexeme "hangs on," find semantically relevant lexeme in the predefined predicate lexicon. Answer: *19.hanging from* |
| **Actual Question** |
| Question: Given the lexeme Input, find semantically relevant lexeme in the predefined predicate lexicon. Answer: |

**SGNLS.** SGNLS employs a pre-trained Faster R-CNN (Ren et al., 2015) object detector trained on Open Images (Kuznetsova et al., 2020) to ground unlocalized triplets. More precisely, it grounds the subject and object within the unlocalized triplet with image regions that share the same class with the subject/object. It is important to note that the set of 601 entity classes in Open Images does not completely cover the 150 entity classes in Visual Genome (Krishna et al., 2017). In other words, there are entity classes in Open Images that do not exist in Visual Genome. Therefore, a knowledge base (i.e., WordNet (Miller, 1995)) is used to align as many of Open Images' entity classes as possible with Visual Genome's entity classes. The aligned entity classes of Open Images are then used to compare with the subjects and objects in the unlocalized triplet for grounding. The predicate within the localized subject and object serves as a pseudo label for training the SGG model.

**VS$^3$.** VS$^3$ employs a grounding-based object detector (i.e., GLIP (Li et al., 2022a)) to ground unlocalized triplets. Specifically, GLIP disentangles the task of object localization, which involves identifying the object's bounding box, and object recognition, which entails recognizing the class associated with that bounding box. Unlike the simultaneous object detection and recognition through Region Proposal Network (RPN) in Faster R-CNN, GLIP initially detects bounding boxes. Then, given the text features corresponding to the entity classes in the target data, it calculates the similarity between these text features (Devlin et al., 2018) and the visual features (Dai et al., 2021) of bounding boxes. The grounding of the subject and object within an unlocalized triplet is achieved by choosing the bounding boxes with the highest similarity scores. Once subject and object grounding is achieved, the predicate serves as a pseudo-label for training the SGG model.

### A.3 DETAILS OF MODEL TRAINING

Here, we provide a detailed explanation for model training after obtaining localized triplets, i.e., $\mathbf{G}_w = \{\mathbf{s}_i, \mathbf{p}_i, \mathbf{o}_i\}_{i=1}^{N_w}$, where $\mathbf{s}_{i,b}$ and $\mathbf{o}_{i,b}$ are obtained from grounding methods. We follow the original training strategy for each method, i.e., SGNLS (Zhong et al., 2021) and VS$^3$ (Zhang et al., 2023).

**SGNLS.** SGNLS uses a Transformer-based Uniter model (Chen et al., 2020) on top of a pre-trained Faster R-CNN (Ren et al., 2015) object detector to capture contextual information of neighboring objects. Each contextualized representation of the subject and object is input into an entity classifier to generate entity logits. The entity classifier and Uniter model are then trained using cross-entropy loss, with supervision provided by entity labels (i.e., $\mathbf{s}_{i,c}$ and $\mathbf{o}_{i,c}$), respectively. For the predicate, its representation is obtained by feed-forwarding the contextualized representations of the subject and object and is fed into the predicate classifier. Then, the predicate classifier and Untier model are trained using cross-entropy loss with supervision on predicate class $\mathbf{p}_{i,c}$.

**VS$^3$.** VS$^3$ builds an additional predicate classifier on top of a pre-trained GLIP (Li et al., 2022a) object detector for predicate prediction. Specifically, the concatenation of visual features and spatial information of $\mathbf{s}_i$ and $\mathbf{o}_i$ is fed into MLP to obtain the predicate representation. Based on its representation, the predicate classifier generates the predicate's logit. The predicate classifier and a cross-modal fusion module within GLIP are trained with supervision provided by predicate class $\mathbf{p}_{i,c}$ using cross-entropy loss. When training entities (subject and object), the approach is similar to the class recognition task in GLIP. Specifically, it maximizes the dot product between the text features of entity classes (i.e., $\mathbf{s}_{i,c}$ and $\mathbf{o}_{i,c}$) and their visual features of bounding boxes using binary focal loss (Lin et al., 2017).

## B REGARDING THE IMPACT OF GROUNDING METHOD ON LLM4SGG

In Table 2 of main paper, we observe that applying LLM4SGG to VS$^3$ (Zhang et al., 2023) (i.e., VS$^3$+LLM4SGG) results in greater performance improvement compared to applying it to SGNLS (Zhong et al., 2021) (i.e., SGNLS+LLM4SGG). We provide a detailed explanation regarding the impact of the grounding method on LLM4SGG.

As mentioned in Section A.2, SGNLS includes the process of aligning the 601 entity classes from the Faster R-CNN (Ren et al., 2015) trained on Open Images (Kuznetsova et al., 2020) with the 150 entity classes in Visual Genome (Krishna et al., 2017). We find that 34 out of 150 entity classes in Visual Genome are not aligned in the end. In other words, the 601 entity classes in Open Images do not cover these 34 entity classes. As a result, unlocalized triplets containing these 34 entity classes are discarded and not used for training since the image regions do not contain the corresponding 34 classes, and they fail to be grounded. In fact, 100K among the 344K unlocalized triplets obtained through LLM4SGG are discarded, exacerbating the low-density scene graph issue. On the other hand, VS$^3$ fully utilizes all 344K triplets. As mentioned in Section A.2, VS$^3$ computes the similarity between text features (Devlin et al., 2018) of entities (subject and object) in the unlocalized triplet and the visual feature (Dai et al., 2021) of each image region. Then, the image region with the highest score is grounded with that entity. This indicates that the subject and object are always successfully grounded with the image regions having the highest score. Therefore, all 344K triplets are being grounded and used for training, effectively alleviating the low-density scene graph issue.

Taking a further step, in Section 3.4, we use LLM4SGG to align the 601 entity classes in Open Images with 150 entity classes in Visual Genome. However, we observe that even if an LLM is used for alignment, 30 entity classes are still not aligned because 30 entity classes have completely different semantic meanings with the 601 entity classes in Open Images. For example, phone and racket do not overlap with the semantic meaning with 601 entity classes in Open Images. This suggests that the grounding method (i.e., SGNLS) with Faster R-CNN trained on Open Images somewhat limits the effectiveness of LLM4SGG.

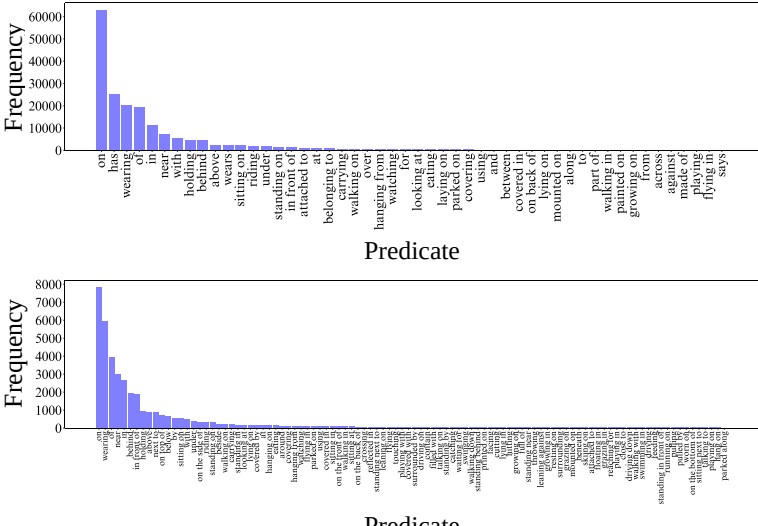

Figure 5: Predicate distribution for Visual Genome (Top) and GQA (Bottom) in test data.

## C EXPERIMENT SETUP

### C.1 DATASET

Visual Genome dataset (Krishna et al., 2017) and GQA dataset (Hudson & Manning, 2019) are widely used in the SGG task. Visual Genome dataset is a benchmark dataset used for evaulating the fully-supervised approach (Yoon et al., 2023; Zhang et al., 2022a; Li et al., 2021b; Khandelwal & Sigal, 2022). For test data, each image has 13.8 objects and 6.9 predicates on average.

Recently, GQA dataset has also been used for evaluating the fully-supervised approach (Li et al., 2023b; Dong et al., 2022; He et al., 2022; Suhail et al., 2021). For test data, each image contains 9.3 objects and 4.6 predicates on average.

The predicate distributions of Visual Genome and GQA are plotted in Figure 5.

### C.2 EVALUATION PROTOCOL

In the SGDet task, a predicted triplet is considered as correct if regions corresponding to the subject and object overlap with the ground-truth boxes with an IoU>0.5, while also having the correct subject and object labels. In the procedure of incorporating the correct triplet into R@K and mR@K performance, we initially compute the triplet score by multiplying the subject score, predicate score, and object score for all subject-object pair in an image, followed by sorting them. If the correct triplet falls within the top-K of the sorted triplets, it contributes to R@K and mR@K performance.

### C.3 BASELINES

For comparison LLM4SGG with baselines, we incorporate the fully-supervised approach (Zellers et al., 2018) and weakly-supervised approach (Ye & Kovashka, 2021; Zhong et al., 2021; Li et al., 2022b; Zhang et al., 2023) (Ye & Kovashka, 2021).

- **Motif** (Zellers et al., 2018) (Fully-supervised): Based on the analysis of repeated patterns (i.e., motif), this method employs Bi-LSTM to capture the motifs appearing across images.
- **LSWS** (Ye & Kovashka, 2021): This method utilizes a Graph Neural Network (GNN) applied to triplets extracted from captions to capture the linguistic structure present among triplets, with the aim of improving grounding unlocalized triplets. Furthermore, this method extends its capability by iteratively estimating scores between image regions and text entities.
- **SGNLS** (Zhong et al., 2021): To ground the unlocalized triplets over image regions, it leverages information from a pre-trained object detector (i.e., Faster R-CNN (Ren et al.,

2015)). Specifically, when the class of a text entity matches a class within a bounding box, the text entity is grounded on that bounding box. Once localized triplets are acquired, a Transformer-based Uniter model (Chen et al., 2020) is trained based on the contextualized representation of entities under the supervision of localized triplets.

- **Li et al. (2022b)**: In the process of grounding unlocalized triplets, this method not only leverages a pre-trained object detector (Ren et al., 2015) for object-aware information but also leverages a pre-trained visual-language model (Li et al., 2021a) to incorporate interaction-aware information.

- **VS³** (Zhang et al., 2023): This method employs a pre-trained object detector (i.e., GLIP (Li et al., 2022a)) to accomplish more than grounding unlocalized triplets; it also aids in identifying novel entities within these triplets. In contrast to earlier WSSGG works that rely on a Faster R-CNN object detector for grounding, this method capitalizes on the grounding ability of the GLIP that disentangles the tasks of class recognition and localization, leading to a significant enhancement of grounding effectiveness.

# D EXPERIMENT ON VG

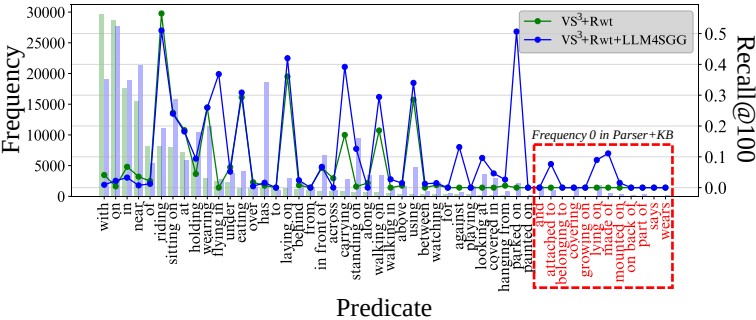

Figure 6: Performance comparison per class when adopting the reweighting method. The red-colored predicates denote the predicates with a frequency of 0 in the conventional approach while LLM4SGG generates all of them with a frequency greater than 0. (Bar: num. predicate instances, Line: Recall@100)

## D.1 PERFORMANCE COMPARISON PER CLASS WITH REWEIGHT METHOD

In Figure 6, we show the performance per class of LLM4SGG adopted to VS³ (i.e., VS³+LLM4SGG) and a baseline (i.e., VS³). We observe that the performance of VS³ + Rwt on 22 fine-grained predicates, which start from the rightmost end of the x-axis *wears* to *between*, drops to nearly zero, although we attempt to enhance the performance of fine-grained predicates with the reweighting method. This is due to the fact that the conventional approach generates unlocalized triplets with a limited number of predicates, and 12 of them even have a frequency of 0. This scarcity of predicates makes it challenging to improve the performance of fine-grained predicates even with reweighting. In contrast, LLM4SGG addresses the semantic over-simplification inherent in the extraction of triplets from captions, which increases the number of fine-grained predicate instances. As a result, when the reweighting method is employed, it effectively boosts the performance of fine-grained predicates. It is worth noting that for some predicates whose frequency is 0 in the conventional approach (i.e., *attached to, lying on, made of, mounted on*), LLM4SGG shows performance improvements, verifying the effectiveness of LLM4SGG.

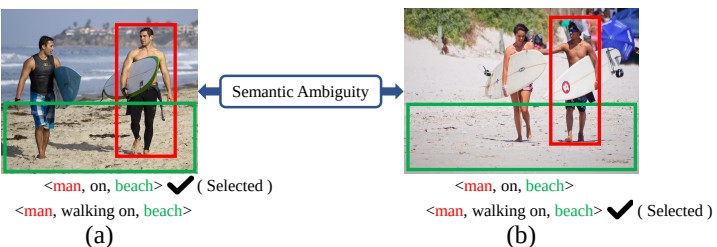

Figure 7: Example of illustrating semantic ambiguity between (a) and (b).

## D.2 Experiment for Predicate Selection

Regarding the implementation details, to further alleviate the long-tailed predicate distribution after Step 3 in our framework, we select the most fine-grained predicate when there are multiple predicates between the same subject-object pair, where the fine-grainedness is determined based on the predicate distribution within the entire set of unlocalized triplets. In Table 9, we further conduct experiments under three different scenarios to understand the effect of predicate selection: random predicate selection, coarse-grained predicate selection, and fine-grained predicate selection. We have the following observations: **1)** LLM4SGG with the random selection (row (a)) and the selection of coarse-grained (row (b)) and fine-grained predicates (row (c)) consistently outperform the baseline with the selection of each case (row (d),(e),(f)), respectively, which verifies the effectiveness of LLM4SGG. **2)** LLM4SGG with a selection of coarse-grained predicates (row (b)) severely deteriorates the mR@K performance while increasing the R@K performance compared to the random selection (row (a)). While R@K performance is improved by increasing the instances of coarse-grained predicates, the fine-grained predicates in an image are not effectively utilized when combined with coarse-grained predicates, resulting in a decrease in the mR@K performance. In contrast, selecting the fine-grained predicates (row (c)) significantly increases the performance of mR@K. **3)** LLM4SGG with a selection of the fine-grained predicates (row (c)) yields higher R@K and mR@K compared to random selection (row (a)). Regarding the improvement of R@K performance, we attribute it to the effect of alleviating the semantic ambiguity (Zhang et al., 2022a). For example, Figure 7(a) and (b) illustrate the two cases where unlocalized triplets parsed from several captions exhibit predicates on and walking on between the same subject man and object beach. If we randomly select the predicates, the model may learn on in one image and walking on in another image even though they are associated with the same subject and object, making the SGG model confused due to the similar visual features with different predicates. For this reason, selecting walking on for both images helps mitigate semantic ambiguity, resulting in enhancing the performance.

Table 9: Experiment for predicate selection

| Row | Selection | R@50 / 100 | mR@50 / 100 | F@50 / 100 |
|-----|-----------|-----------|-------------|------------|
| | | VS$^3$+LLM4SGG | | |
| (a) | Random | 8.15 / 9.55 | 5.10 / 6.19 | 6.27 / 7.51 |
| (b) | Coarse-grained | **9.79 / 11.37** | 2.52 / 3.03 | 4.01 / 4.78 |
| (c) | Fine-grained | 8.91 / 10.43 | **7.11 / 8.18** | **7.91 / 9.17** |
| | | VS$^3$ | | |
| (d) | Random | 6.60 / 8.01 | 2.88 / 3.25 | 4.01 / 4.62 |
| (e) | Coarse-grained | **6.99 / 8.20** | 2.66 / 2.99 | 3.85 / 4.38 |
| (f) | Fine-grained | 6.18 / 7.43 | **3.82 / 4.27** | **4.72 / 5.42** |

## D.3 Experiment for Exploration of Training Space

In Table 10, we show the results of zero-shot Recall@K (zR@K), as introduced by (Tang et al., 2020), to explore the training space of triplets extracted from captions. It is important to note that the zR@K metric used in the fully-supervised approach evaluates how well the model predicts $\langle$subject, predicate, object$\rangle$ triplet sets that have never been observed in the training data of Visual Genome dataset (Krishna et al., 2017). Therefore, while triplets extracted from the captions in COCO dataset may accidentally overlap with these zero-shot triplets, we employ this metric to understand how much our proposed approach broadens the training space. When we compare VS$^3$, which learns from triplets extracted from captions through the conventional approach, to Motif, a fully-supervised approach trained on Visual Genome dataset, we observe that VS$^3$ achieves a higher zR@K, implying that captions contain a broader range of diverse compositional relationships compared to the Visual Genome dataset. On the other hand, VS$^3$ with LLM4SGG (i.e., VS$^3$+LLM4SGG) significantly improves the zR@K performance compared to VS$^3$. This suggests that triplets generated through LLM4SGG are proficient at capturing the compositional relationships found in captions, thereby expanding the training space of triplets. This expansion is achieved by addressing the semantic over-simplification issue, leading to the creation of a more varied set of predicates, and the low-density scene graph, resulting in a wider range of compositional triplets. We argue that the use of the zR@K metric demonstrates the effectiveness of LLM4SGG in terms of expanding the training space.

Table 10: Performance comparison for exploration of training space.

| Method | zR@50 | zR@100 |
|--------|-------|--------|
| Motif (Fully-supervised) | 0.31 | 0.60 |
| VS$^3$ | 1.16 | 1.46 |
| VS$^3$+LLM4SGG | **2.20** | **3.02** |

## D.4 Experiment for New Prompt Design

In Table 11, we conduct an experiment with a new prompt design. In fact, the prompt for extracting triplets from captions in Section 3.3 and the alignment of entity/predicate classes with target data in Section 3.4 can be combined into

Table 11: Experiment for new prompt.

| Method | R@50 / 100 | mR@50 /100 | F@50 / 100 |
|---|---|---|---|
| VS³ | 6.60 / 8.01 | 2.88 / 3.25 | 4.01 / 4.62 |
| VS³+LLM4SGG$_{comb}$ | 8.66 / 10.20 | 6.28 / 7.06 | 7.28 / 8.34 |
| VS³+LLM4SGG | **8.91 / 10.43** | **7.11 / 8.18** | **7.91 / 9.17** |

one. More precisely, we instruct the LLM to follow the four steps for triplet extraction from paraphrased caption: Step 1) Paraphrase the caption, Step 2) Extract meaningful triplets from the paraphrased caption obtained in Step 1, Step 3) Find semantically relevant lexeme in the predefined entity lexicon for the subject and object from the triplets obtained in Step 2, where the entity lexicon is enumerated in Table 7. Step 4) Find semantically relevant lexeme in the predefined predicate lexicon for the predicate from the triplets obtained in Step 3, where the predicate lexicon is enumerated in Table 8. Following the four steps, we include stepwise results in the in-context examples for in-context few-shot learning, and insert the caption from which we want to extract triplets in the actual question. As shown in Table 11, LLM4SGG with the combined prompt (i.e., VS³+LLM4SGG$_{comb}$) outperforms the baseline, implying that even though prompt design can be diverse, it consistently verifies the efficacy of the LLM-based triplet formation process. On the other hand, VS³+LLM4SGG$_{comb}$ shows inferior performance compared to VS³+LLM4SGG. This is due to a practical reason incurred by the length limit in the prompt of GPT-3.5, which allows only up to 4096 tokens. Specifically, VS³+LLM4SGG$_{comb}$ integrates the instructions of all four steps in the task description, including the definition of entity/predicate lexicons, and in-context examples following four steps within a single prompt, which leads to a longer length while the maximum length is constrained. For this reason, we could only accommodate four in-context examples. In contrast, VS³+LLM4SGG divides the four steps into two chains (Section 3.3 and Section 3.4), allowing more in-context examples in each chain. Specifically, VS³+LLM4SGG contains six in-context examples in Chain-1 (i.e., triplet extraction) and fourteen in-context examples in Chain-2 (i.e., alignment of entity/predicate classes). The increased number of in-context examples equips an LLM to adapt more effectively to the provided few-shot task (Wei et al., 2022b), ultimately enhancing its adaptability for the task of triplet extraction task and alignment of entity/predicate classes. In summary, under the practical length limit of GPT-3.5, we observe that an approach that divides the original four steps into two chains is more effective in extracting triplets aligned with entity/predicate of interest than an approach that combines them into a single prompt.

### D.5 CASE STUDIES ON EXTRACTING FINE-GRAINED PREDICATES

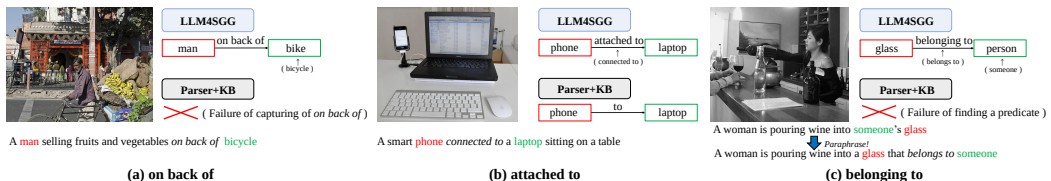

Figure 8: Case studies on extracting fine-grained predicates with a frequency of 0 in the conventional approach (i.e., Parser+KB). (a): LLM4SGG extracts the fine-grained predicate found in the caption. (b): LLM4SGG extracts the fine-grained predicate while aligning it with a predicate in the target data. (c): In LLM4SGG, paraphrasing the caption via LLM aids in identifying a fine-grained predicate, imparting a more specific meaning to it.

In Figure 8, we present case studies on predicates in which the conventional approach (i.e., Parser+KB) eventually ends up in a frequency of 0, while LLM4SGG does not. In Figure 8(a), despite the presence of the fine-grained predicate on back of in the caption, the conventional approach fails to extract it since the conventional approach relies on a heuristic rule-based parser (Wu et al., 2019) that lacks the capability of understanding the context of caption and extracting the predicate on back of at once. On the other hand, LLM4SGG successfully extracts the predicate on back of through a comprehensive understanding of the caption's context. In Figure 8(b), we observe that the conventional approach, which suffers from semantic over-simplification, extracts the coarse-grained predicate to instead of connected to, whereas LLM4SGG extracts the fined-grained predicate connected to within the caption. Subsequently, by aligning connected to with semantically relevant lexeme, attached to, in the target data through LLM's semantic reasoning ability, it eventually generates a fine-grained predicate attached to. This demonstrates the effec-

tiveness of LLM-based triplet extraction in addition to LLM-based alignment, leading to capturing the fine-grained predicates. Interestingly, in Figure 8(c), we observe that the paraphrasing step (Section 3.3) in LLM4SGG aids in extracting fine-grained predicates. Specifically, paraphrasing the caption conveys a more specific meaning, i.e., from *someone's glass* to *glass that belongs to someone*, thus enabling the extraction of the fine-grained predicate belonging to, which the conventional approach cannot achieve. Through these case studies, we demonstrate the effectiveness of extracting the fine-grained predicates in LLM4SGG.

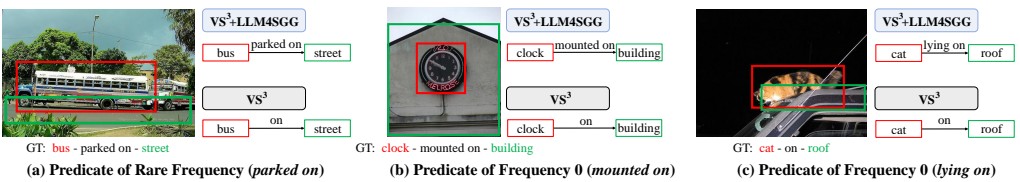

(a) Predicate of Rare Frequency (*parked on*)    (b) Predicate of Frequency 0 (*mounted on*)    (c) Predicate of Frequency 0 (*lying on*)

Figure 9: Qualitative results on Visual Genome dataset. (a) A predicate *parked on* rarely appears in the conventional approach. (b), (c) Predicates *mounted on* and *lying on* never appear in the conventional approach.

### D.6 QUALITATIVE RESULTS

To further verify the effectiveness of LLM4SGG on Visual Genome dataset (Krishna et al., 2017), in Figure 9 , we showcase qualitative results from the test data, comparing the baseline (i.e., VS$^3$ (Zhang et al., 2023)) with LLM4SGG applied to VS$^3$ (i.e., VS$^3$+LLM4SGG) . In Figure 9(a), we observe that LLM4SGG accurately predicts a fine-grained predicate parked on between subject bus and object street, which rarely appears in the training data using the conventional approach. In contrast, the baseline (i.e., VS$^3$) following the conventional approach predicts a coarse-grained predicate on due to the semantic over-simplification, incurring the long-tailed predicate distribution. Furthermore, in Figure 9(b), we observe that LLM4SGG makes a correct prediction on the predicate whose frequency is 0 in the conventional approach (i.e., mounted on). On the other hand, the baseline predicts a coarse-grained predicate on and never predicts mounted on since it has not been observed during training. Interestingly, while in Figure 9(c) LLM4SGG made an incorrect prediction by predicting lying on instead of on, we argue that lying on is a more realistic answer that provides a richer context. However, as lying on is never observed while training the baseline (i.e., VS$^3$), it can never be predicted. Through the qualitative analysis, we again demonstrate the effectiveness of alleviating the long-tailed problem in WSSGG.

## E EXPERIMENT ON GQA

### E.1 TRAINING AND EVALUATION

**Training.** To train a SGG model for evaluation on the GQA dataset (Hudson & Manning, 2019), LLM4SGG requires three modifications, encompassing the change of 1) predefined entity, 2) predicate lexicon, and 3) in-context examples in Section 3.4, while maintaining the triplet extraction process in Section 3.3. More precisely, in the predefined entity and predicate lexicons, we replace the entity and predicate classes originally associated with Visual Genome dataset (Krishna et al., 2017) with entity and predicate classes in the GQA dataset, respectively. For in-context examples, we substitute the examples that are initially relevant to Visaul Genome dataset with examples related to GQA dataset. After obtaining unlocalized triplets composed of entity/predicate classes in the GQA dataset, the process of grounding them remains unchanged from the grounding method used in Visual Genome dataset. Please refer to the details of the grounding process in Section A.2.

**Evaluation.** To evaluate a trained model on the GQA dataset, only a single modification is needed in the architecture of VS$^3$ (Zhang et al., 2023). Specifically, we change the output entity class of bounding boxes from the entity classes of the Visual Genome dataset to those of the GQA dataset. For details of VS$^3$ regarding the determination of entity classes for bounding boxes, the target data's entity classes are listed in text format, e.g., airplane. animal. arm. ... . Then, a text encoder (Devlin et al., 2018) encodes the enumerated text list to obtain the text features for each entity. The similarity

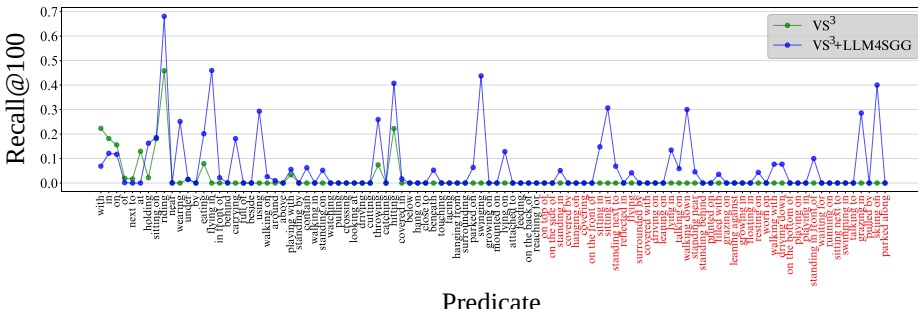

Figure 10: Performance comparison per class on GQA dataset. The red-colored predicates indicate that the frequency of predicate instances generated by the conventional approach is 0, while there are no predicates with a frequency of 0 in LLM4SGG.

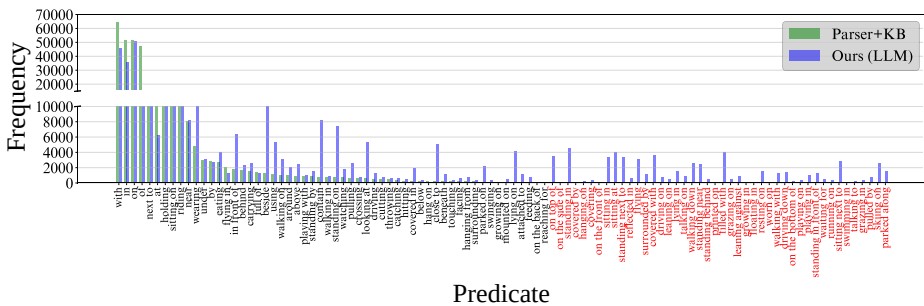

Figure 11: Predicate distribution on GQA dataset. The red-colored predicates indicate that the frequency of predicate instances generated by the conventional approach is 0, while there are no predicates with a frequency of 0 in LLM4SGG.

between these text features of entity classes and visual features (Dai et al., 2021) of the bounding boxes is computed. In perspective of the bounding box, the text entity with the highest similarity is chosen as the class assigned to the bounding box. In the procedure of determining the entity classes for bounding boxes, we simply list the entity classes from the GQA dataset instead of Visual Genome's entity classes. This ensures that entity classes assigned to the bounding boxes align with the entity classes in the GQA dataset. Subsequently, we proceed with a standard evaluation protocol used in Visual Genome dataset.

### E.2 PERFORMANCE COMPARISON PER CLASS

In Figure 10, we show a performance comparison per class on the GQA dataset (Hudson & Manning, 2019). For baseline (i.e., VS$^3$ (Zhang et al., 2023)), we observe that the performance for most predicates, except for coarse-grained predicates, is nearly zero. It is attributed to the semantic over-simplification inherent in the conventional approach, leading to a long-tailed predicate class distribution as shown in Figure 11. The long-tailed problem is severely exacerbated in the GQA dataset (Hudson & Manning, 2019) due to its inclusion of more complicated predicates (e.g., standing next to), as well as a greater variety of fine-grained predicates, such as talking on and driving down, which are challenging to extract from captions using heuristic rule-based parser (Wu et al., 2019). In fact, 44 out of 100 predicates have a frequency of 0, meaning that they are never predicted. On the other hand, as shown in Figure 11, LLM4SGG effectively addresses the long-tailed problem by alleviating the semantic over-simplification and low-density scene graph issues, increasing the instances that belong to the fine-grained predicates instances so that there are no predicates with a frequency of 0. As a result, LLM4SGG significantly enhances the performance of fine-grained predicates, thereby improving the performance of mR@K. This demonstrates the effectiveness of LLM4SGG with the more challenging GQA dataset.

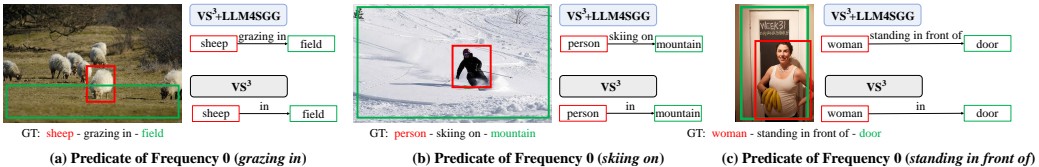

GT: sheep - grazing in - field

GT: person - skiing on - mountain

GT: woman - standing in front of - door

**(a) Predicate of Frequency 0 (*grazing in*)**   **(b) Predicate of Frequency 0 (*skiing on*)**   **(c) Predicate of Frequency 0 (*standing in front of*)**

Figure 12: Qualitative results on GQA dataset. (a), (b), (c): predicates *grazing in*, *skiing on*, and *standing in front of* never appear while training the baseline VS[3].

### E.3 QUALITATIVE RESULTS

To further demonstrate the effectiveness of LLM4SGG on a more challenging dataset, GQA (Hudson & Manning, 2019), we present qualitative results comparing the baseline (i.e., VS[3]) with LLM4SGG applied to VS[3] (i.e., VS[3]+LLM4SGG) in Figure 12. For Figure 12(a), (b), and (c), we showcase examples from the test data, where predicates have a frequency of 0 in the training data when triplets are generated using the conventional approach, i.e., grazing in, skiing on, and standing in front of. In Figure 12(a) and (b), we observe that the baseline predicts the coarse-grained predicate in, which is the second most frequent predicate, as shown in Figure 11. The prediction of coarse-grained predicate is attributed to the semantic over-simplification issue of the conventional approach, leading to a long-tailed problem. On the other hand, LLM4SGG correctly predicts the fine-grained predicates grazing in and skiing on by effectively alleviating the semantic over-simplification issue. In Figure 12(c), we encounter a complicated predicate, standing in front of, between the subject woman and object door. Such predicates are intricate to extract from captions unless they are comprehensively understood and extracted at once. LLM4SGG, however, adeptly extracts the fine-grained predicate standing in front of by understanding the entire context of the caption via LLM. LLM4SGG makes it possible to learn the predicate standing in front of, resulting in a correct prediction. These qualitative results on the more challenging GQA dataset further verify the effectiveness of LLM4SGG.

