# OpenReview forum: "Weakly Supervised Fine-grained Scene Graph Generation via Large Language Model"
_ICLR.cc/2024/Conference — ICLR 2024 Conference Withdrawn Submission_

### Official Review · Reviewer_mv1X · 2023-10-28

**Soundness:** 3 good
**Presentation:** 3 good
**Contribution:** 2 fair
**Rating:** 5
**Confidence:** 4

**Summary:**

Weakly-supervised scene graph generation suffers from low quality data -- most studies extract <subject, predicte, object> triplet from image captions yet captions provide incomplete representation or biased (e.g., coarse-grained predicates) information. This paper proposed a way to augment the training data to improve the quality of the training <subject, predicte, object> triplets. Experiments show that augmenting training data using LLM improved the final WSSGG performance.

**Strengths:**

The paper is well presented and easy to follow. The proposed approach targets the painpoint of the current WSSGG methods -- they focus too much on the modeling yet ignored the data quality in triplets extraction. Experiments of the paper are concrete and showing the positive improvements.

**Weaknesses:**

Literature study: summarization of the knowledge base completion (a NLP task) is totally missing from the paper's related work section. The KB-completion requires to fill-in-the-blank for the missing information in a knowledge graph, while from my point of view the paper's main contribution is to use LLM to complete the information extracted from the caption.

Novelty: applying LLM to complete the knowledge graph is not new, and chain-of-thought prompting is new neither.

Comparison: the proposed approach augments the training dataset to improve model's performance on tailed distributed triplets. Since training data are modified, the authors should propose a new benchmark, I'd suggest they open source the LLM-augmented data for future studies.

Missing a quantitative study of the quality of the augmented data.

Lacking modeling contribution & possibility of improving the language modeling methodology: prior works such as Ye & Kovashka, 2021 and Zhong et al., 2021 have included language module, so I wonder the performance of replacing their language module with a LLM. For example, Ye & Kovashka, 2021 mentioned that "plate-on-pizza" is not making sense thus their language module correct it to "pizza-on-plate". It seems to me that LLM can also be used to correct the knowledge. So, I'd suggest an additional study to use LLM to refine model outputs instead of changing the training data.

**Questions:**

Suggestions:
Adding summarization of the  knowledge base completion works will make the literature review more complete.
Adding quantitative study of the quality of the augmented data would strengthen the paper and explain the improvement.
If the LLM can be used to replace the LM in the prior works, the contribution of the paper is more sound.

---

### Official Review · Reviewer_XUZu · 2023-10-30

**Soundness:** 3 good
**Presentation:** 2 fair
**Contribution:** 2 fair
**Rating:** 5
**Confidence:** 4

**Summary:**

This paper focuses on the weakly-supervised scene graph generation task. Existing image caption-based WSSGG methods inevitably overlook two issues: (1) Semantic over-simplification issue. (2) Low-density scene graph issue. Therefore, a new approach named Large Language Model for weakly-supervised SGG (LLM4SGG) is proposed in this paper. LLM4SGG utilizes the in-depth understanding of language and reasoning ability of LLMs for the extraction of triplets from captions and the alignment of entity/predicate classes with target data. Extensive experiments on the VG dataset and the GQA dataset demonstrate the effectiveness of the proposed LLM4SGG.

**Strengths:**

+ Two major issues overlooked by existing WSSGG studies are identified in this paper, i.e., semantic oversimplification and low-density scene graph.
+ This paper leverages an LLM along with the CoT strategy and the in-context few-shot learning technique to extract informative triplets without fine-tuning the LLM. It is the first to leverage an LLM for the SGG task.
+ LLM4SGG outperforms the state-of-the-art WSSGG methods, especially regarding mR@K, demonstrating its efficacy in addressing the long-tail problem in WSSGG for the first time.

**Weaknesses:**

- As the authors stated, "To the best of our knowledge, we are the first to leverage an LLM for the SGG task". However, there are some works that employ LLM for SGG, such as [a].
- The method proposed in this paper is indeed a way of generating more appropriate and precise triplets by utilizing LLMs. Considering that this paper only relies on the generative capacity of LLM to augment the dataset, it is not enough to prove its novelty. In addition, it is well known to all that prompts play a crucial role in harnessing the potential of LLMs. So how to construct or select suitable prompts for your tasks? Do you have any experimental results or explanations for the construction of prompts? Besides, some implementation details about LLMs should be introduced in the paper, such as the number of GPUs used.
- In addition to the approaches utilizing image captions as the weak supervision, there are other weakly-supervised approaches that adopt graph matching for alignment (using graphs as weak supervision). Further comparative analysis is needed. Can the proposed method be used in the graph-based weakly-supervised SGG?
-(Minor) More explanations are needed for the phenomenon that the number of triplets per image in the COCO caption dataset is less than that in the VG dataset. The conflict between the static structured knowledge of the knowledge base (KB) and covering semantic relationships among a wide range of words remains ambiguous; more exemplifications are required.
-(Minor) In this paper, there are still some errors or mistakes in the typesetting of the figures that need to be addressed, such as the lack of content in the orange boxes in step 1 and step 2 of Figure 1.

[a] Zero-shot Visual Relation Detection via Composite Visual Cues from Large Language Models, arXiv 2023.
[b] Shi J, Zhong Y, Xu N, et al. A simple baseline for weakly-supervised scene graph generation[C]//Proceedings of the IEEE/CVF International Conference on Computer Vision. 2021: 16393-16402.

**Questions:**

Please refer to the weakness.

---

### Official Review · Reviewer_rSZy · 2023-11-01

**Soundness:** 3 good
**Presentation:** 3 good
**Contribution:** 2 fair
**Rating:** 5
**Confidence:** 4

**Summary:**

The paper introduces LLM4SGG, a method that uses a large language model to generate weakly supervised scene graph data. By leveraging an LLM in the predicate extraction and alignment procedure, the method can better extracts finer-grained predicates and achieve better data efficiency by preserving more aligned data. It empirically shows performance improvements over baseline approaches.

**Strengths:**

- The method is intuitive and using an LLM in the weakly supervised scene graph data generation pipeline is reasonable given their powerful language understanding capabilities.
- It leads to improvements empirically especially on the long-tail predicates.
- The method is more data efficient, achieving similar/better performance to the baseline when using less training data.

**Weaknesses:**

- The method is rather straightforward, by replacing conventional parser with LLMs, and leveraging LLMs more like blackbox language processing tools.
- The method relies on proprietary blackbox LLMs. It is not clear if the method is robust to work with smaller language models.
- It has been observed in recent literature that LLMs may have problem processing long input texts [1]. The alignment step also requires the LLM to identify semantically relevant lexeme from a potentially large set of predefined lexicon. It is not clear how well the method could scale when the lexicon grows.

[1] Lost in the Middle: How Language Models Use Long Contexts. Liu et al. 2023.

**Questions:**

- Could the authors provide an estimate on the cost associating with querying ChatGPT in generating the weakly supervised scene graph?
- Figure 3 is a bit small and hard to read.

---

### Official Review · Reviewer_EXTF · 2023-11-01

**Soundness:** 3 good
**Presentation:** 3 good
**Contribution:** 3 good
**Rating:** 6
**Confidence:** 5

**Summary:**

This paper works on the weakly-supervised scene graph generation (WSSGG). Specifically, they argue that the most popular caption-based WSSGG has overlooked two questions: semantic over-simplification and low-density scene graph. For the semantic over-simplification problem, they think the rule-based parser with heuristic rules falls short of accommodating the diverse range of the caption's structure. For the low-density scene graph problem, existing WSSGG methods suffer from the lack of sufficient supervision per image, leading to poor generalization and performance degradation. To this end, they propose a new approach LLM4SGG, which adopts a pretrained LLM for WSSGG. For the first problem, they directly use LLM to extract triplet instead of rule-based parser. For the second problem, they ask LLM to paraphrase the original caption and extract more triplets from the paraphrased caption. Then, they use LLM to further align them with semantically relevant lexeme. Extensive results have shown that LLM4SGG significantly enhances the performance of existing WSSGG methods on both Visual Genome and GQA datasets.

**Strengths:**

+ The motivation is very clear, ie, existing caption-based WSSGG methods suffer from these two issues, and it would be helpful if the mode can help to mitigate both issues.

+ The proposed method is general enough. It would consistently improve the performance of multiple existing baselines.

**Weaknesses:**

+ Since all techniques rely on the advanced ability of LLM, the errors in LLMs will be propagated to the following stages.

**Questions:**

+ Since the key ideas aim to improve the quality of pseudo scene graph annotations for fully supervised SGG models training, it would be better to show the reasons for performance improvement, such as more and dense annotations.

+ Based on the results in Table 2, the VS^3+Rwt+LLM4SGG achieves better performance than the fully-supervised method Motifs on mean Recall metrics. It would be better to have more explanations about this.